# Birnessite: A New Oxidant for Green Rust Formation

**DOI:** 10.3390/ma13173777

**Published:** 2020-08-26

**Authors:** Amira Doggaz, Romain Coustel, Pierrick Durand, François Humbert, Christian Ruby

**Affiliations:** 1CNRS, LCPME, Université de Lorraine, 54600 Nancy, France; amira.doggaz@univ-lorraine.fr (A.D.); romain.coustel@univ-lorraine.fr (R.C.); francois.humbert@univ-lorraine.fr (F.H.); 2CNRS, CRM2, Université de Lorraine, 54600 Nancy, France; pierrick.durand@univ-lorraine.fr

**Keywords:** layered double hydroxide, LDH, manganese, iron, fougerite, oxidation

## Abstract

Iron and manganese are ubiquitous in the natural environment. Fe^II^-Fe^III^ layered double hydroxide, commonly called green rust (*GR*), and Mn^III^-Mn^IV^ birnessite (*Bir*) are also well known to be reactive solid compounds. Therefore, studying the chemical interactions between Fe and Mn species could contribute to understanding the interactions between their respective biogeochemical cycles. Moreover, ferromanganese solid compounds are potentially interesting materials for water treatment. Here, a {Fe(OH)_2_, Fe^II^_aq_} mixture was oxidized by *Bir* in sulphated aqueous media in the presence or absence of dissolved O_2_. In oxic conditions for an initial Fe^II^/OH^−^ ratio of 0.6, a single *GR* phase was obtained in a first step; the oxidation kinetics being faster than without *Bir*. In a second step, *GR* was oxidised into various final products, mainly in a spinel structure. A partial substitution of Fe by Mn species was suspected in both *GR* and the spinel. In anoxic condition, *GR* was also observed but other by-products were concomitantly formed. All the oxidation products were characterized by XRD, XPS, and Mössbauer spectroscopy. Hence, oxidation of Fe^II^ species by *Bir* can be considered as a new chemical pathway for producing ferromanganese spinels. Furthermore, these results suggest that *Bir* may participate in the formation of *GR* minerals.

## 1. Introduction

Manganese (Mn) oxides minerals are commonly present in a wide range of natural environments such as ocean nodules, soil coatings and sediments [1]. Although a variety of these minerals exists, birnessite-type manganese oxides are the most widely studied due to their wide distribution, chemical reactivity and redox properties. Birnessite exhibited excellent oxidation capacity for organic molecules [2], inorganic complexes [3] and toxic metallic ions such as As(III) [4], Cr(III) [5], and Co(II) [6]. Birnessite was also shown to participate in the oxidation of ferrous ions and Fe(II) containing minerals [7,8,9,10]. The relation between the kinetics of Fe^2+^_aq_ oxidation and the presence of birnessite was investigated in aqueous solutions under anoxic condition [8,10]. Reduction of Mn(IV) by Fe^2+^_aq_ led to Fe(III) solid compounds and the influence of pH was studied using synthetic birnessite Mn_7_O_13_.H_2_O [10]. The reaction was slower at pH above 4 due to the precipitation of FeOOH on the birnessite surface that blocked reactive sites. In presence of oxygen, the redox reaction between Fe^2+^_aq_ and hexagonal birnessite (K_0.23_MnO_2.03_·0.6H_2_O) was boosted at pH 5.5 in comparison with the reaction in anoxic environment [8]. This was attributed to the regeneration of Mn(IV) due to the fact that the newly formed Mn(III) species generated by the reduction of Mn(IV) in birnessite would be re-oxidized by air [11,12]. The same overall mechanism was also observed in the case of the oxidation of pyrite [7]. Birnessite affected also the crystallization processes and the nature of the iron oxide precipitate [9]. The precipitation product formed after Fe^2+^ oxidation at pH 6.0 was found to be lepidocrocite γ-FeOOH in the absence of birnessite. By increasing the Mn/Fe ratio from 0.01 to 1, the precipitate was found to be either a mixture of lepidocrocite and goethite or a non-crystalline iron(III) phase.

Most of the previous studies focused on the oxidation of Fe^2+^_aq_ ions by birnessite in acidic medium. However, literature provides little insight on the interaction process between birnessite and solids such as iron(II) hydroxides. Moreover, to the best of our knowledge, the reactivity with iron of a triclinic birnessite compound initially synthesized in alkaline conditions was never studied. The oxidation of aqueous suspensions of Fe(II) species by various oxidants, such as air and persulfate, was reported [13,14]. Nevertheless, no attempt was made to study the influence of a solid oxidant such as birnessite on the oxidation process of Fe(II) hydroxide. The oxidation of Fe(II) hydroxide led generally to either magnetite Fe_3_O_4_ or ferric oxyhydroxides FeOOH, but in specific conditions mixed Fe(II)-Fe(III) hydroxyde, also called green rust (GR), was identified [15,16,17,18]. GR is a layered double hydroxide composed by Fe(II) and Fe(III) hydroxide units intercalated with water molecules and ions such as sulfate, chloride, and carbonate ions. The general formula was established to be [$Fe_{1-x}^{II}Fe_{x}^{III}{\left(OH\right)}_{2}{]}^{x+}{\left[{\left(A\right)}_{\frac{x}{n}}.yH_{2}O\right]}^{x-}$, where A^n−^ is an anion, *x* varied between ~0.25 and 0.33 and *y* was situated in between ~0.5 and ~1.33 [14]. GR was synthesized either by coprecipitation of Fe^2+^_aq_ and Fe^3+^_aq_ species [14] or by the oxidation of {Fe^2+^_aq_, Fe(OH)_2_} mixtures [15]. The reaction of oxidation was controlled by various parameters such as concentration of the reactants, temperature, stirring speed, nature of the Fe(II) salt and depended strongly upon the ratio *R* = [Fe^2+^]/[OH^−^] of initial suspension [15,16,17,18]. Moreover, depending on all these conditions, further oxidation of GR resulted in the formation of different kinds of iron oxides with different structures and oxidation states: magnetite, lepidocrocite, goethite or hematite [15]. For *R* = 0.5, i.e., Fe(OH)_2_ stoichiometry, the oxidation product was essentially magnetite Fe_3_O_4_ [15]. In the presence of an excess of Fe^2+^ ions, and more specifically for *R* = 0.6, two successive equilibriums took place during the oxidation process: the equilibrium between Fe(OH)_2 (s)_ and GR in a first step, and the equilibrium between GR and FeOOH_(s)_ in a second step [17]. The exact value R = 0.6 ratio was determined to be the appropriate ratio to obtain solely hydroxysulfate GR according to a reaction independent of the pH [17]:(1)$$5{\mathrm{Fe}}^{\mathrm{II}}{\left(\mathrm{OH}\right)}_{2}+{\;\mathrm{SO}}_{4}^{2-}+{\mathrm{Fe}}^{2+}+\frac{1}{2}O_{2}+H_{2}O↔{\;\mathrm{Fe}}_{4\;}^{\mathrm{II}}{\mathrm{Fe}}_{2}^{\mathrm{III}}{\left(\mathrm{OH}\right)}_{12}{\mathrm{SO}}_{4}$$

This paper aims to highlight the influence of birnessite on Fe(II) hydroxide oxidation process in sulfate containing medium. Experiments were carried out using conditions allowing GR formation in oxic condition. In a first step, the iron (II) hydroxide-birnessite interaction was studied either in anoxic condition or in an aerated medium. In a second step, the oxidation reaction was investigated in presence of Fe^2+^ ions in solutions in order to evaluate the role birnessite on GR formation. Careful attention was devoted to determining the status of both Fe and Mn species during the formation of GR and its final oxidation products by using redox potential measurements, X-ray diffraction, X-ray photoelectron spectroscopy, and Mössbauer spectrometry.

## 2. Materials and Methods 

### 2.1. Synthesis of Birnessite

Triclinic Na-birnessite was synthetized using the “alkali method” based on a redox reaction between MnO_4_^−^ and Mn^2+^ in alkaline conditions [19]. Subsequently, 250 mL of 0.1 mol L^−1^ NaMnO_4_ solution was mixed quickly with 125 mL of MnCl_2_ (0.6 mol L^−1^) under vigorous stirring. Then, 125 mL of 8.8 mol L^−1^ NaOH was added dropwise during 2 h. The mixture reaction was then stirred for another 30 min and aged at 60 °C for 14 h. The product was finally centrifuged and washed until the pH of the solution was 10 and dried in an oven at 60 °C for 16 h. The average oxidation state of Mn in birnessite was determined to be 3.5, and the chemical formula was written as Na_0.35_MnO_2_H_2_O_0.7_ [19].

### 2.2. Oxidation of Ferrous Species by Air and Birnessite

Ferrous hydroxide precipitate was prepared by mixing 80 mL of an aqueous solution of melanterite FeSO_4_·7H_2_O and 20 mL of NaOH solution in a beaker. For The ratio *R* = [Fe^2+^]/[OH^−^] of 0.5 (0.6, respectively), FeSO_4_·7H_2_0 and NaOH concentrations were fixed at 0.1 mol L^−1^ and 0.2 mol L^−1^ (0.12 mol L^−1^ and 0.2 mol L^−1^, respectively). The ratio of 0.6 provided an excess of dissolved Fe^2+^_aq_. The suspension was aerated at the interface between solution and atmosphere by magnetic stirring to ensure a progressive oxidation of the suspension at constant velocity. The redox potential (*E*_h_) monitored by using an Ag/AgCl electrode and the pH was measured continuously in the suspension. In a first series of experiments Fe(OH)_2_ was oxidized by air with or without presence of birnessite (0, 2 and 5 g L^−1^) and in a second series only by birnessite under nitrogen blanket (to prevent oxidation by O_2_). In the end of the run, filtration was conducted before solid characterization. Samples were denoted (RXX)(BYY)(Z) (see Table 1) with *XX* = 05 or 06 when *R* = 0.5 or 0.6, *YY* = 02 or 05 if oxidation was performed with birnessite concentration of 2 or 5 g L^−1^ and Z = I, F or IN2 for Intermediate product, Final product obtained under atmospheric conditions or Intermediate product obtained under N_2_ blanketing, respectively. The formation of intermediate or final products is detailed in Section 3.1.1. 

### 2.3. Solids Characterization 

#### 2.3.1. X-ray Diffraction (XRD)

XRD measurements were performed using a Panalytical X’Pert Pro diffractometer equipped with a Cu tube, a Ge(111) incident-beam monochromator (Kα1 = 1.5406 Å), 0.02 rad Soller slits, programmable divergence, and anti-scatter slits. The irradiated area was fixed to 10 mm × 10 mm. A X’Celerator detector was used as “scanning line detector (1D)” with 2.122° active length. The intermediate products were characterized soon after the synthesis and centrifugation. As the compounds contain water and are known to be air sensitive, excess of water was removed by pressing the powder between tissues. The resulting powder (paste) was characterized on a 200 μm thick zero background X-ray holder covered with a minimum amount of glycerol to protect the powder from air oxidation. Further experimental details are given in Appendix B. 

#### 2.3.2. X-ray Photoelectron Spectroscopy (XPS) 

X-ray photoelectron spectra were recorded on a KRATOS Axis Ultra X-ray photoelectron spectrometer with Al Kα source monochromated at 1486.6 eV (spot size 0.7 mm × 0.3 mm). Photoelectrons were detected by a hemispherical analyzer at an electron emission angle of 90° and pass energy of 20 eV (core level spectra). For the core-level spectra, the overall energy resolution, resulting from monochromator and electron analyzer bandwidths, was 800 meV. As an internal reference for the absolute binding energies, the C1s peak of hydrocarbon contamination set at 284.6 eV was used.

#### 2.3.3. Transmission Mössbauer Spectrometry

^57^Fe Mössbauer spectra were collected using a spectrometer in transmission geometry coupled with a cold head cryostat from Janis Research, equipped with vibration isolation stand, developed in LCPME laboratory. The 50 mCi ^57^Co(Rh) source was driven with constant acceleration. Measurements were taken in the range ± 11 mm s^−1^. The hyperfine interaction parameters were determined by fitting the experimental spectra by a least-squares method using the Recoil software [20]. The center shifts were reported with respect to that of 25 μm-thick α-Fe foil at room temperature.

## 3. Results

### 3.1. Air Oxidation of Fe^II^ Species in the Presence of Birnessite 

#### 3.1.1. Oxidation of Fe(OH)_2_ without Fe^2+^_aq_ (*R* = 0.5)

Fe(OH)_2_ suspension prepared with R = 0.5 is essentially Fe^2+^_aq_ free. Its oxidation by both ambient oxygen and birnessite (0, 2 or 5 g L^−1^) was investigated recording pH and potential E_h_ evolution over time (Figure 1). In the reference experiment, without birnessite, pH decreased during the first 50 min and then stabilized. Therefore, in the experiments with birnessite, the latter was added after 50 min of the run in order to highlight the role of birnessite in the oxidation process. Suspensions containing birnessite were more alkaline than the reference without birnessite. For all the experiments, E_h_ values were almost stable (−550 mV/SHE) until *t* = 210 min. When no birnessite was added, this plateau corresponds to the equilibrium between Fe(OH)_2_ and magnetite (Fe_3_O_4_). Beyond 220 ± 10 min, both pH and *E*_h_ values exhibited significant variations indicating the breakdown of the initial equilibrium and the full oxidation of Fe(OH)_2_ into magnetite.

XRD characterization of solids present after 240 min (samples R05F, R05B05F) showed that the final product of the reaction was magnetite for all the studied systems (Figure 2 and Table 1). 

Therefore, the following reaction of formation occurred: (2)$$3{\;\mathrm{Fe}}^{\mathrm{II}}{\left(\mathrm{OH}\right)}_{2}+\frac{1}{2}{\;O}_{2}\to {\mathrm{Fe}}^{\mathrm{II}}{\mathrm{Fe}}_{2}^{\mathrm{III}}O_{4}+\;3{\;H}_{2}O$$

Whatever the birnessite amount, the plateau length remained constant showing that the presence of birnessite did not modify the oxidation kinetics (Figure 1).

#### 3.1.2. Oxidation of a {Fe(OH)_2_, Fe^2+^_aq_} Mixture (*R* = 0.6)

Experiments were performed using similar experimental conditions as for the runs performed for R = 0.5, except for the initial value of R fixed here at 0.6. It should be noted also that birnessite was added as soon as Fe^2+^/OH^−^ mixture was prepared. As shown in Figure 1, the reaction time required to reach the end of the first E_h_ plateau (equilibrium GR/end product) was strongly dependent on the dose of birnessite. The time corresponding to the end of the first E_h_ plateau measured without birnessite was 315 min while it did not exceed 190 and 110 min for 2 and 5 g·L^−1^ of birnessite, respectively. The solid products corresponding to the samples withdrawn at the end of the first E_h_ plateau are called intermediated products (samples R06I, R06B02I and R06B05I). Their XRD patterns evidenced hydroxysulfate green rust [18] as major species and were observed independently of the birnessite dose (see Figure 3 and Table 1). In the absence of oxygen, the xrd pattern of solid recovered after about 110 min (R06B05IN2 sample) also evidenced hydroxysulfate green rust as major species (Appendix A and Table 1). Note that the diffractogram of the intermediated products (Figure 3 and Appendix A) shows wide hump in the 15°–25° region typical of amorphous phase. This feature is due to the presence of glycerol that prevents product oxidation during XRD measurements for at least one night.

Note that a small increase of the cell parameter *a* from 3.17 to 3.19 Å was measured for products obtained with increasing birnessite amount (see Table 1). This suggests a partial substitution of Fe species by Mn species inside the GR structure. The potential substitution of Fe species by Mn species inside the oxidation products will be discussed in detail later (Section 4.3). Without birnessite, the *E*_h_ curve in Figure 1 shows two equilibrium plateaus corresponding respectively to the transformation of Fe(OH)_2_ into GR and to the subsequent oxidation of GR into magnetite. XRD patterns of the intermediate product at the end of the first plateau (sample R06I Figure 3) and of the final product after 410 min reaction time (sample R06F Figure 2) sustained this mechanism. In the presence of birnessite, the length of the E_h_ first plateau was strongly reduced and the end of the second plateau (*E*_h_ ~ −300 mV/SHE) corresponding to the oxidation of GR into magnetite was clearly observed (Figure 1). 

### 3.2. Reduction of Birnessite Followed by XRD and XPS

Birnessite phase was no more evidenced by powder XRD measurements in R06B02F and R06B05F samples (Figure 2). Indeed the low angle and most intense diffraction peak of birnessite situated at 2θ ~12.5 ° disappeared completely suggesting a full reduction reaction of birnessite with Fe(II) species after 400 min in oxygenated conditions. Moreover, hausmannite identification by powder XRD as a minor phase in R06B05F (Figure 3 and Table 1) was consistent with the birnessite reduction. To go further into the understanding of the fate of Mn, additional XPS measurements were carried out on R06B02F and R06B05F samples. Actually, XPS is an effective tool to discriminate Mn(II) from Mn(III) and Mn(IV) in solid compound [19,21].

Overview XPS spectra (not shown) showed core-level photoelectron peaks at ~285 eV (C1s), ~530 eV (O1s), ~642 eV (Mn2p3/2), ~654 eV (Mn2p1/2), ~710 eV (Fe2p3/2), and ~725 eV (Fe2p1/2). The C1s peak was attributed to atmospheric hydrocarbon contamination of the sample surface before introduction into the XPS chamber. Amounts of S and Na were also identified from features at ~168 eV (S2p), ~230 eV (S2s), and 1070 eV (Na1s).

The Mn2p 3/2 XPS spectra of R06B02F and R06B05F samples and as prepared birnessite were compared to the spectrum of as prepared birnessite (Figure 4). 

The maximum of the peak shifts from 642.0 eV for birnessite to 641.4 eV for R06B05F and to 641.0 eV for R06B02F. Shift to lower binding energy indicated a diminution of the average oxidation state at sample surface. The decomposition of Mn 2p 3/2 peaks according to Nesbitt et al. recommendations [21] sustained this analysis. In this procedure, each contribution of Mn(II), Mn(III), and Mn(IV) was modelled as a multiplet structure whose parameters were derived from ab initio calculations. Birnessite spectrum was fitted with Mn(IV) and Mn(III) contributions (area ratio equal to 43:57, respectively). Mn(IV) contribution tends to vanish for both R06B02F and R06B05F samples while a significant Mn(II) contribution gave rise to a shoulder at ~640 eV. Interestingly, the R06B02F spectrum presented also a more or less pronounced contribution at 646 eV generally attributed to a shake-up peak of Mn(II) species, for instance in MnO [21]. Therefore, XPS measurements corroborated that manganese in birnessite was reduced by the {Fe(OH)_2_; Fe^2+aq^} mixture.

## 4. Discussion

### 4.1. Thermodynamical Analysis of the E_h_-pH Data 

As described in previous studies [17,22], the E_h_ and pH plateau values recorded during the Fe^II^ oxidation process were used to determine the standard chemical potential of hydroxysulfate green rust (GRSO_4_). A value µ°(GRSO_4_) = −3790 ± 10 kJ mol^−1^ was determined [17] in agreement with other works [23]. In fully aerated sulfated aqueous medium, the initial {Fe^2+^_aq_, Fe(OH)_2_) mixture was in a first step transformed into GRSO_4_ (first E_h_ and pH plateaus) and in a second step GRSO_4_ was transformed into lepidocrocite γ-FeOOH (second E_h_ and pH plateaus). The E_h_ values of both plateaus were related to the Nernst electrochemical equilibria of the GRSO_4_/ Fe(OH)_2_ and γ-FeOOH / GR redox couples, respectively. 

In the experiment performed without birnessite, the formation of GRSO_4_ ended after an oxidation time *t*_1_ ~320 min., a value that is significantly higher than the one recorded in similar studies performed at comparable Fe^II^ concentration (~10^−1^ M), e.g., *t*_1_ was measured to be 50 min [13], 90 min [16], and 100 min [17] in previous works. Therefore, the aeration rate of the aqueous medium was almost three times lower in our study in comparison to the previous studies. By knowing the standard chemical potential of the different species involved in the oxidation process, the values of *E*_h_ plateau was used as an indicator to determine the nature of the various oxidation products. The *E*_h_= −530 ± 5 mV/SHE and pH = 8 ± 0.05 values recorded at the middle of the first *E*_h-_pH plateau (Figure 1) are comparable to the redox potential measured in other studies, e.g., *E*_h_ = –497 mV/SHE and pH = 8 [17] and *E_h_* ~510 mV /SHE and pH ~8.3 [16]. The slightly lower E_h_ value recorded in our study may be related to different reasons: (i) the stability of the reference electrode used for the experiment, i.e., either saturated calomel [16,17] or Ag/AgCl in this study, (ii) a slight difference of the initial conditions (concentrations of sulfate anions and eventual presence of chloride anions in the solution [16], (iii) the aeration condition as discussed in previous works [17,22]. Therefore, the E_h_ values of the first plateau were globally in agreement with the formation of GRS0_4_ as already observed with XRD (Figure 3). Note that the lower oxidation kinetics of the {Fe(OH)_2_, Fe^2+^_aq_} mixture in comparison to previous studies [17,22] did not modify the nature of intermediate oxidation product, i.e., GRSO_4_, at the end of the first plateau. 

In the presence of birnessite the E_h_ values recorded during the first plateau did not vary significantly despite the fact that a significant increase of pH was measured (Figure 1). An explanation is obtained by considering the Fe(OH)_2_/GRSO_4_ electrochemical equilibrium {Equation (3)} and the corresponding Nernst Equation (4).
6Fe(OH)_2_ + SO_4_^2−^ = Fe^II^_4_Fe^III^_2_(OH)_12_SO_4_ + 2 e^−^(3)
*E*_h_ (GRSO_4_/Fe(OH)_2_) = *E*_h_° (GRSO_4_/Fe(OH)_2_) − 0.0295 log{*a*(SO_4_^2−^)}(4)

As observed in Equation (4), the redox potential *E*_h_(GRSO_4_/Fe(OH)_2_) does not depend on the pH. Therefore, the electrochemical equilibrium between Fe(OH)_2_ and GRSO_4_ was maintained in the presence of birnessite and the increase of pH did not induce any *E*_h_ variation.

The *E*_h_ = −295 ± 5 mV/SHE value measured for the experiment performed without birnessite at the middle of the second plateau was situated at significant lower values than those recorded previously, i.e., *E*_h_ = −75 mV/SHE when GRSO_4_ was transformed into lepidocrocite γ-FeOOH [17]. The oxidation reactions of GRSO_4_ into either lepidocrocite γ-FeOOH or magnetite Fe_3_O_4_ and the corresponding Nernst equations are given in Appendix C. The equilibrium potentials E_h_(γ-FeOOH/GRSO_4_) ~−120 mV and E_h_(Fe_3_O_4_/ GRSO_4_) ~−462 mV were computed by using Equations (A1) and (A3), respectively, and a pH of 6.8 measured after a reaction time of 400 min. (Figure 1). Considering the activity coefficient of sulfate species as performed in other works will have a very low influence on the computed values of *E*_h_ [17]. The redox potential *E_h_* = −295 ± 5 mV/SHE measured experimentally for the R06F sample was clearly situated in between both previous values in good agreement with the formation of a mixture of magnetite and ferric oxyhydroxides as observed by XRD (Figure 2 and Table 1). A slightly lower value of the redox potential mean value (*E_h_* ~−325 ± 10 mV/SHE) was measured for the second E_h_ plateau for the experiment performed with 2 and 5 g L^−1^ of birnessite. For a concentration of 5 g L^−1^ of birnessite, the second plateau almost disappeared showing that GRSO_4_ was quickly oxidized into magnetite by the excess of birnessite (see the minor phase R06B05I sample in Table 1) present in the aqueous medium at the end of the first *E*_h_ plateau.

### 4.2. Kinetics of the Oxidation Reaction

Oxidation of dissolved Fe(II) by Mn(III-IV) oxides is thermodynamically favorable and is well documented [10,24]. In order to understand the interactions between Fe^2+^_aq_ and hexagonal birnessite, the reaction was studied in previous works in acidic solution since both hexagonal birnessite and Fe^2+^_aq_ were stable in the initial suspension. Postma studied the kinetics and stoichiometry of the reaction between Fe^2+^_aq_ and synthetic birnessite from pH 3 to 6 at 10 °C [10]. The kinetics was shown to be fast at pH < 4. The reaction proposed involved the reduction of Mn(IV) and the oxidation of Fe^2+^_aq_ according to the following reaction [10,25]: (5)$${\mathrm{Mn}}_{7}O_{13}5H_{2}O+\;12{\mathrm{Fe}}^{2+}+26H^{+}\to \;7{\mathrm{Mn}}^{2+}+\;12{\mathrm{Fe}}^{3+}+18H_{2}O$$

At higher pH, the reaction was initially fast and then quickly slowed due to the precipitation of the Fe^3+^ species into Fe(III) oxyhydroxides, probably on the birnessite surface. The overall redox reaction was supported later by Gao et al. suggesting an adsorption of Fe^2+^ on birnessite before oxidation [8]. In aerobic environment, Fe^2+^ removal rate and Mn^2+^ release significantly increased, it was proposed that oxygen may regenerate the Mn(IV) leading to an acceleration of the oxidation reaction due to a catalytic effect. The reaction kinetics in unbuffered solution under anoxic conditions was also investigated by using a polymeric MnO_2_ compound [24]. The reaction was determined to be of first order with respect to MnO_2_ and Fe^2+^ and of the second order overall. Similar to the work of Gao et al. [8], Mn(III) intermediates were produced during the reaction.

In the present study, a birnessite with a triclinic structure was synthesized in alkaline solutions as already observed in a former work [19]. The choice to work with triclinic birnessite was also guided by the fact that green rust is generally formed under neutral or alkaline conditions [17]. On one hand both oxygen and birnessite participate in the green rust formation from suspension with *R* = 0.6 in aerated condition. On the other hand, no reaction between Fe(OH)_2_ and birnessite was evidenced in suspension with *R* = 0.5. It should be noted that the latter R ratio led to iron into Fe(OH)_2_ without any free Fe^2+^_aq_ in solution. Hence, the acceleration of the oxidation reaction leading to GR by birnessite with *R* = 0.6 can only be explained by the presence of dissolved Fe^2+^_aq_ in the medium. Moreover, birnessite showed a capacity to oxidize the {Fe^2+^_aq_, Fe(OH)_2(s)_} mixture even in the absence of oxygen (sample R06B05IN2 in Table 1). According to the redox reaction and to the chemical formula Mn_7_O_13_.H_2_O of birnessite [10], the quantity of oxidant was calculated to be in excess in comparison with the quantity of Fe(II) species in the initial suspension (see details in Appendix C), in good agreement with the excess of birnessite observed by XRD for sample R06B05I (Table 1). 

Several researchers focused on studying the birnessite role on the precipitation products of iron [8,9]. The nature of precipitation products of Fe varied depending on the pH and the Mn/Fe molar ratio [9]. The final oxidation products of Fe range from lepidocrocite through goethite, akaganeite to noncrystalline Fe oxides by increasing Mn/Fe ratio. However, in these previous studies, no intermediate mixed Fe(II)-Fe(III) compounds were observed during the oxidation process. In this work the formation Fe(II)-Fe(III) green rust was observed before further oxidation into the final oxidation products, mainly a spinel structure with ferric oxyhydroxydes as minor by-products. As will be discussed in the following section, the partial substitution of Fe by Mn species in both green rust and the spinel is suspected. 

### 4.3. Potential Substitution of Fe by Mn in the Oxidation Products

#### 4.3.1. Powder X-ray Diffraction Analysis

As already mentioned, powder XRD measurements showed a very slight and progressive increase of the *a* cell parameter of sulphated green rust for increasing amount of birnessite in the initial suspension (Table 1). The exact value of the cell parameters *a* = 3.1776(3) Å (equivalent to the supercell *a* = 5.504 Å) recorded for the R06I sample was in good agreement with the reported values, i.e., either *a*√3 = 5.524 Å [18] or *a*√3 = 5.507 Å recorded using high resolution powder XRD [26]. Such a superstructure was also visible in the diffractogram recorded in this study (see Figure 3): the superstructure gave a series of very low intensity diffraction peaks with a main one situated around 2θ = 18.6°. One should note that the most intense line of magnetite was located at 35.4°: such a contribution of the (311) family planes (ICSD: 01-089-2355) was identified for R06I and R06B05I. The (111) family planes of magnetite gave a ~10 times less intense contribution at 18.28°. The line observed at 18.6° was situated at a significantly higher angle position, while its intensity was similar or stronger than that of the 35.4° line. Therefore, the 18.6° line was assigned to green rust.

The *c* parameter recorded here (*c* = 10.9638 Å) was also in perfect agreement with the one reported previously for sulphated green rust, *c* = 10.9664 Å [26]. The progressive increase of the cell parameter *a* proportionally to the initial birnessite amount can be related to a partial substitution of some of the Fe species by Mn species inside the metallic brucite-type sheet of green rust. Indeed, the effective high spin (HS) ionic radius of both Mn(II) and Mn(III) (82 and 65 pm) in oxides is higher than the same radius for Fe(II) and Fe(III) species (63 and 49 pm). Such a difference may explain the observed global expansion of the cell. Moreover, the difference of electronegativity between HS Mn(II) and HS-Fe(II) calculated according to the scale of Li and Xue [27] was very low. This is another indication that Fe(II) species may be easily substituted by Mn(II) species. The interlayer distance *c* depends strongly on the size of the sulphated species that form a double layer [18] and on the electrostatic charge of the layers. However, the *c* parameter is not very sensitive to the chemical nature of the cations. The substitution of Fe by Mn species of identical charge, e.g., Fe(II) by Mn(II), does not modify the electrostatic charge of the layer in agreement with the invariability of the *c* parameter. Moreover, a continuous broadening of the XRD diffraction peaks was observed when the R06I diffractogram was compared to those of R06B02I and R06B05I samples. Therefore, the partial substitution of Fe by Mn species may also induce lower crystallinity, so a decrease of the crystallite size of green rust. The question arises whether such a substitution would also occur in the major final oxidation products, i.e., the spinel structure similar to the one of magnetite.

Single phase of magnetite Fe_3_O_4_ was observed by XRD for sample R05F (Figure 2 and Table 1). Tick marks of theoretical patterns of magnetite and hausmannite were also shown as guidelines in Figure 2. Rietveld refinements were performed on the pure magnetite sample using as a starting structural model of magnetite or maghemite. The lattice constants, profile coefficients and background fit were refined, the chemical nature of the atoms being fixed. Magnetite and maghemite have very close X- ray diffractograms, but the Rietveld analysis clearly showed that the lattice parameter *a* obtained (~8.37 Å) was situated in between those published for magnetite (~8.40 Å) and for maghemite (~8.34 Å) [28].

LeBail extractions were performed on the other final oxidation products containing both the spinel structure and extra phases (R06B05F, R06F, R05B05F samples). A subtle but significant lattice expansion was measured, in particular for the R06B05F sample with a value of *a* close to 8.41 Å (see Appendix A). This last result suggests that magnetite, similarly to green rust, would be partially substituted by Mn as higher *a* cell parameters were observed in MnFe_2_O_4_ (*a* ~ 8.47 Å) in comparison with Fe_3_O_4_ (*a* ~ 8.40 Å). Finally, though the intensity of the (111) peak at about 18.3° is almost null for the maghemite structure, it remains significant for magnetite (Fe_3_O_4_) as well as for MnFe_2_O_4_, which is in agreement with the measured data. Mössbauer spectrometric measurements were carried out to get a further insight into both the characterization of the final oxidation products and the potential substitution of Fe by Mn species in the spinel structure.

#### 4.3.2. Analysis with Mössbauer Spectroscopy

The ^57^Fe Mössbauer spectra recorded at room temperature on R06F, R06B02F, and R06B05F are shown in Figure 5 and the corresponding hyperfine parameters are given in Table 2.

In line with XRD data (Table 1), the Mössbauer spectrum of R06F appeared as the sum of several contributions (Figure 5a). Two minor ferric doublets with Δ= 1.19 mm s^−1^ and 0.55 mm s^−1^ were distinguished. The hyperfine parameters of the former corresponded very well to ferric sulfate like natrojarosite [29]. The latter was tentatively attributed to lepidocrocite that was also suspected from XRD data (Table 1). A broad sextet whose hyperfine field distribution maximum was located at 35.5 T (see inset in Figure 5) and was be attributed to goethite. Finally, the main contribution to the spectrum was fitted with three sextets (S1, S2, and S3) and was ascribed to magnetite. Stoichiometric magnetite Fe_3_O_4_ formula was written as [Fe^3+^]_A_[Fe^2+^Fe^3+^]_B_O_4_ where A and B denoted for tetrahedral and octahedral sites, respectively. The iron ions in the octahedral sites led to fast electron hopping and an averaged valence state Fe^2.5+^ [30]. S1 with CS = 0.24 mm s^−1^, <H_hf_> = 48.9 T and small ε was assigned to high spin Fe^3+^ in tetrahedral site and S2 with CS = 0.66 mm s^−1^, <H_hf_> = 45.4 T and also small ε to mixed valence Fe^2.5+^ in octahedral site. These values were in good agreement with results obtained for bulk magnetite [30]. The supplementary sextet S3 with CS = 0.42 mm s^−1^ and <H_hf_> = 49.5 T was assigned to Fe^3+^ in octahedral site as expected for non-stoichiometric magnetite with formula Fe_3-δ_O_4_ ≡[Fe^3+^]_A_[Fe^2.5+^_2(1-3δ)_Fe^3+^_5δ_ □_δ_]_B_O_4_ (δ = 0.33 for maghemite Fe_2_O_3,_ [31], □ stands for vacancy). From the relative area (RA) of S1, S2, and S3, the deviation from ideal stoichiometry δ was determined according to the recommendations of Zegeye et al. [32]. The obtained δ value for R06F was equal to 0.13 indicating that magnetite was partially oxidized.

The Mössbauer spectrum of R06B02F presented paramagnetic and magnetic components attributed to residual nonmagnetic ferric oxyhydroxide and ferrite phase, respectively. As for R06F, the magnetic component was fitted with three sextets (see Table 2). However, the weakening of the hyperfine field, the broadening of the sextets as well as the weakening of [Fe^2.5+^]_B_ contribution with respect to R06F supported that obtained ferrite was partially substituted with Mn [33,34,35]. XPS measurements on R06B02F evidenced Mn^II^ and Mn^III^. Tetrahedral and octahedral sites of partially oxidized magnetite can easily accommodate these species [36,37]. The complete description of the resulting iron environments is beyond the scope of this work.

The Mössbauer spectrum of R06B05F showed paramagnetic and magnetic components in equal RA. The magnetic component was fitted with two sextets with CS equal to 0.29 mm s^−1^ and 0.42 mm s^−1^, indicating that iron is in ferric state. The average hyperfine fields of the sextets were also significantly reduced relative to stoichiometric magnetite and maghemite [31] and even relative to the R06B02F sample. Such reduction may be due to (i) higher Mn substitution rate into the obtained ferrite (expected from higher birnessite/Fe ratio) that weakens Fe-Fe interaction [33,34,35] as well as to (ii) poorer crystallinity leading to weaken long-range interaction. Actually, XRD lines broadening evidenced poorly crystallized ferrite in R06B05F (Figure 2). The paramagnetic component (CS = 0.33 mm s^−1^ and <Δ> = 0.66 mm s^−1^) might be attributed to nanoparticles in superparamagnetic state. It is worth to note that MnFe_2_O_4_ nanoparticles were reported to give rise to paramagnetic component with similar hyperfine parameters at room temperature (CS = 0.33 mm s^−1^ and <Δ> = 0.69 mm s^−1^ [38]. The doublet progressively changed into broad singlet at low temperature (see Appendix A). This result sustained the superparamagnetic behavior of R06B05F even if, in the absence of complete magnetic splitting at 77 K, the contribution of additional ferric phase to the central pattern was not excluded.

Finally, Mössbauer spectra analysis indicated that the CS of the [Fe^2.5+^]_B_ component was reduced from 0.66 mm s^−1^ in R06F to 0.51 mm s^−1^ in R06B02F, while R06B05F spectrum did not present any contribution whose CS was higher than 0.42 mm s^−1^ (see Table 2). This trend was in line with the diminution of Fe(II) amount in the series R06F, R06B02F and R06B05F confirming the role of birnessite as oxidizing agent of ferrous species. This result was consistent with Mn reduction evidenced by XPS as well as the global increasing value of the final E_h_ values (Figure 1).

## 5. Conclusions

Triclinic birnessite prepared in alkaline conditions was able to accelerate the formation of sulfated green rust for R = 0.6 in aerated condition. Green rust was an intermediate Fe(II)-Fe(III) oxidation product that further oxidized into several phases, mainly into a spinel structure in slightly aerated conditions. The refined analysis performed with powder XRD and Mössbauer spectroscopy suggested a partial substitution of Fe species by Mn species in both green rust and the final spinel structure. Further work could be devoted to better understanding the status of Mn and Fe in such a spinel structure for a better control of ferromanganese materials synthesis. Moreover, in anoxic conditions, birnessite reacted as a unique oxidant with {Fe(OH)_2_, Fe^II^_aq_} to give green rust. Like air or persulfate, birnessite appears to be a new oxidant for green rust synthesis. Finally, a redox reaction between Mn and Fe species may play a role in the formation of green rust, in particular at the frontier between the oxic and anoxic zones of hydromorphic soils.

## Figures and Tables

**Figure 1 materials-13-03777-f001:**
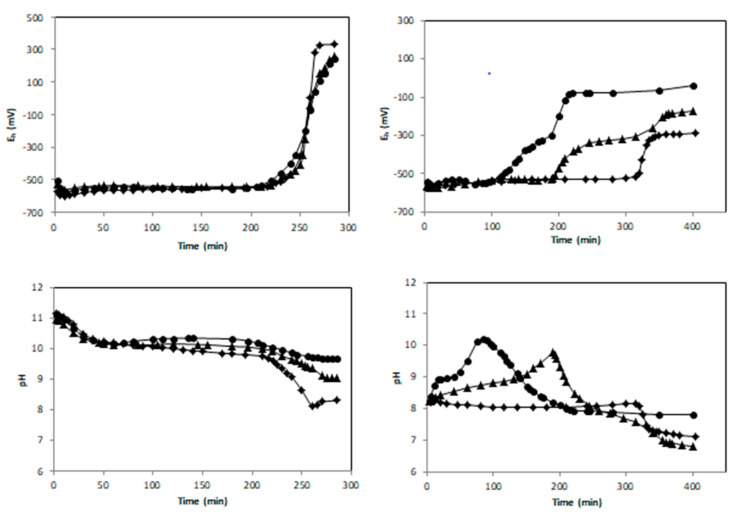
*E*_h_ and pH vs time for *R* = 0.5 (**left panel**, birnessite added at *t* = 50 min) and *R* = 0.6 (**right panel**, birnessite added at *t* = 3 min): diamond, triangle and circle correspond to [Bir] = 0, 2 and 5 g·L^−1^, respectively.

**Figure 2 materials-13-03777-f002:**
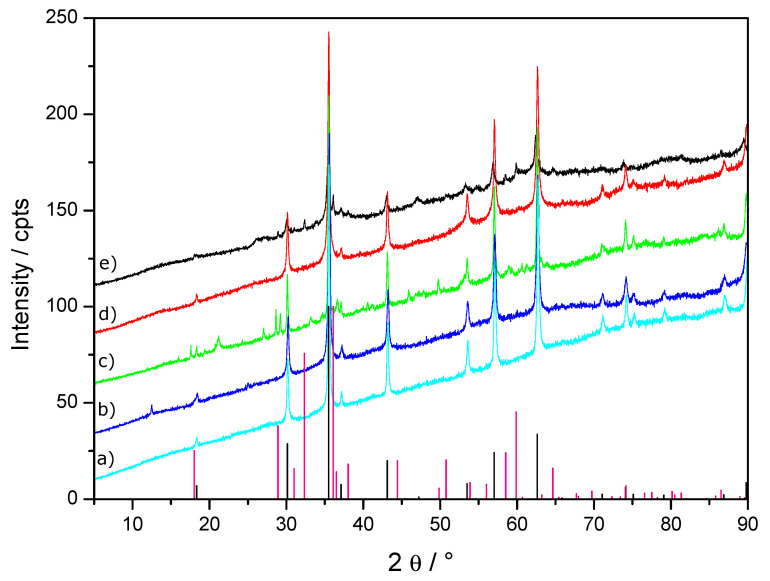
XRD patterns of (**a**) R05F (cyan), (**b**) R05B05F (blue), (**c**) R06F (green), (**d**) R06B02F (red), (**e**) R06B05F (black) samples. Tick marks of the patterns of magnetite (black, ICSD: 01-075-0033) and hausmannite (grey, ICSD: 01-089-4837) are also shown as guidelines.

**Figure 3 materials-13-03777-f003:**
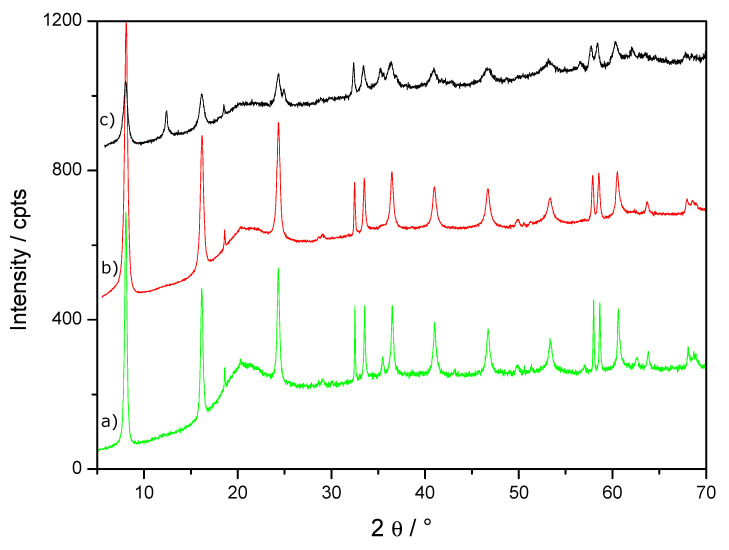
XRD patterns of (**a**) R06I (green), (**b**) R06B02I (red), (**c**) R06B05I (black) samples.

**Figure 4 materials-13-03777-f004:**
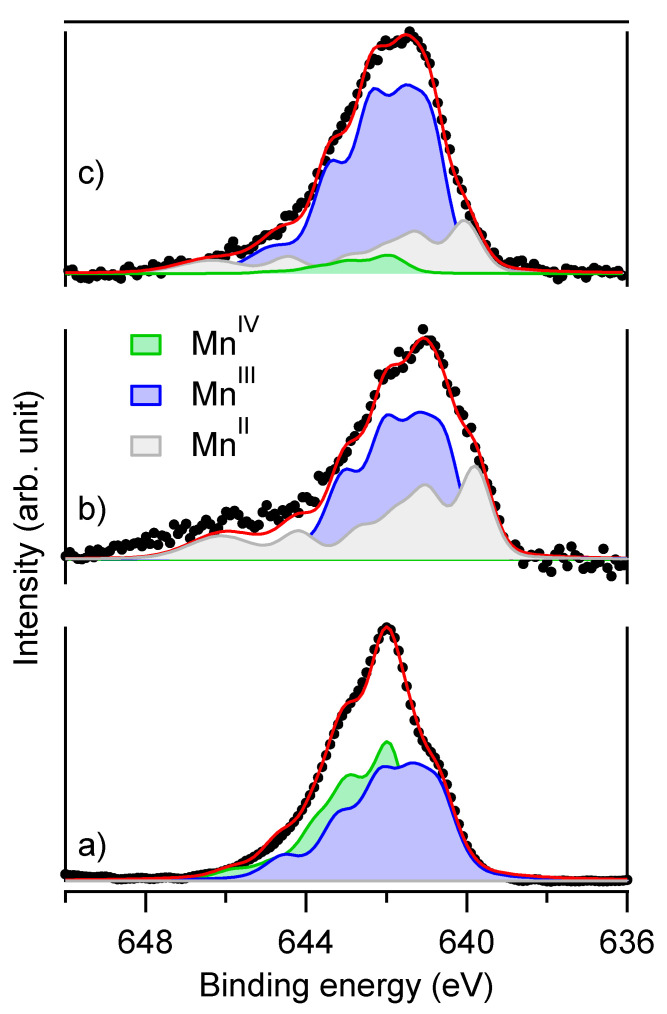
Mn2p 3/2 XPS spectra of (**a**) as prepared birnessite, (**b**) R06B02F and (**c**) R06B05F.

**Figure 5 materials-13-03777-f005:**
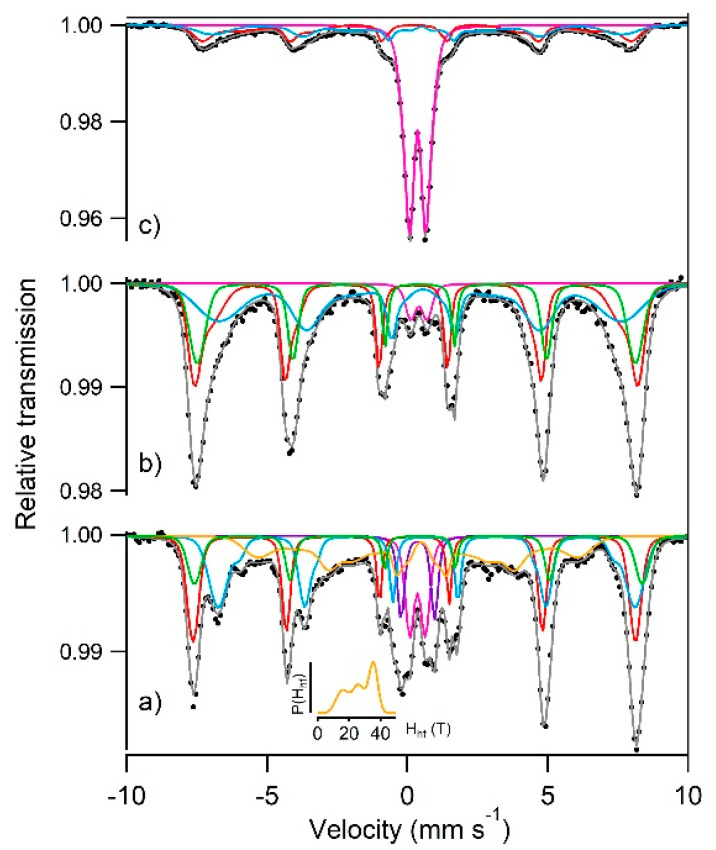
^57^Fe Mössbauer spectra of (**a**) R06F, (**b**) R06B02F, (**c**) R06B05F recorded at room temperature. The inset in (a) gives the probability density distribution of the sextet S4.

**Table 1 materials-13-03777-t001:** Major and minor solid compounds identified by XRD during the oxidation process of Fe(II) species either in the absence or in the presence of birnessite.

	Sample	Major Phase		Minor Phases
			Cell Parameters (Å)	
Final Products	R06B05F	Magnetite	*a* = 8.412 (3)	Hausmannite Mn_3_O_4_ Suspected: Manganite MnO(OH) Suspected: Lepidocrocite FeO(OH)
	R06B02F		*a* = 8.3814(5)	None
	R06F		*a* = 8.3828(11)	Goethite FeO(OH) Natrojarosite NaFe_3_(SO_4_)_2_(OH)_6_ Suspected: Lepidocrocite FeO(OH)
	R05B05F		*a* = 8.3846(12)	Na Birnessite
	R05F		*a* = 8.3722(5)	None
Intermediate Products	R06I	Green rust	*a* = 3.1776(3) *c* = 10.9638(23)	Magnetite Fe_3_O_4_
	R06B02I		*a* = 3.1841(12) *c* = 10.976(7)	None
	R06B05I		*a* = 3.1912(17) *c* = 10.973(9)	Na Birnessite Magnetite Fe_3_O_4_
	R06B05IN2		*a* = 3.1874(16) *c* = 10.954(8)	Na Birnessite Hausmannite Mn_3_O_4_ Magnetite Fe_3_O_4_

**Table 2 materials-13-03777-t002:** Mössbauer hyperfine parameters for the spectra of R06, R06B02 and R06B05 recorded at room temperature (see Figure 5).

Sample	Site	CS (mm s^−1^)	<Δ> or ε (mm s^−1^)	<H_hf_> (T)	RA (%)	
R06F	D1	0.36	0.56	-	9	Lepidocrocite
-	D2	0.37	1.20	-	6	Natrojarosite
-	S1	0.24	0.00	48.9	23	Fe_3-__δ_O_4_ [Fe^3+^]_A_
-	S2	0.66	0.02	45.4	26	Fe_3-__δ_O_4_ [Fe^2.5+^]_B_
-	S3	0.42	−0.03	49.5	12	Fe_3-__δ_O_4_ [Fe^3+^]_B_
-	S4	0.43	−0.06	27.2	24	Goethite
-	-	-	-	-	-	-
R06B02F	D	0.40	0.58	-	4	Lepidocrocite
-	S1	0.26	0.05	47.5	37	Mn_x_Fe_3-x-__δ_O_4_ [Fe^3+^]_A_
-	S2	0.51	−0.06	39.9	37	Mn_x_Fe_3-x-__δ_O_4_ [Fe^2.5+^]_B_
-	S3	0.38	−0.06	48.4	23	Mn_x_Fe_3-x-__δ_O_4_ [Fe^3+^]_B_
-	-	-	-	-	-	-
R06B05F	D	0.36	0.66	-	53	-
-	S1	0.29	0.04	43.7	24	-
-	S2	0.42	-0.08	35.0	23	-

CS: center shift relative to alpha-iron, <Δ>: average quadrupole splitting of paramagnetic component, ε: quadrupole shift, <Hhf> average hyperfine field, RA: relative abundance of each site.

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
