# Peer review of "Birnessite: A New Oxidant for Green Rust Formation"

_materials, 2020, doi:10.3390/ma13173777_

Round 1

Reviewer 1 Report

The paper deals with a widely treated topic in the literature, so a high novelty level is not present. In addition, at the end of the paper, the authors only perform hypotheses about a possible mechanism and not clear conclusions are provided. This can be due to the choice of not suitable experimental technique to demonstrate the partial Fe substitution by Mn ions.

Some other points are the following:

  • XRD patterns have an increasing background. Why? I know that the background level is high due to the iron presence, but such a form is strange
  • I don’t understand the necessity to invoke a superstructure from the peak at about 18.6° (line 359) that is typical of magnetite
  • SI 2: in the R06B05IN2 pattern reflections from magnetite (about 18.5 and 35°) were neglected.
  • Mossbauer analysis does provide only suggestion of the presence of some kind of phases.
  • The authors, to clarify the phases present in the XRD patterns (bulk technique) use the XPS technique (a surface one). It should be possible and also expected that different results can be obtained!

Due to the presentation of only hypotheses and not a complete and definitive description of the mechanism of involved oxidation, I think that the paper does not provide any meaningful advancement in the corresponding topic. In addition, the choice of experimental technique to demonstrate the hypotheses is not right.

The paper should be rejected for the publication in Materials

Reviewer 2 Report

The subject of the paper is interesting for an average number of readers, especially those working in the field of inorganic chemistry. However the paper is well written and it deserves publication after the authors solve the following issues:

1) page 5 top - the authors should give some explanation for the the loss of the equilibrium after 220 min

2) for the data presented in Fig 1, right panel R=0.6 at 240 min there is a very poor equilibrium state (in the pH value) which lasts for no more than 30 minutes for [Bir] of 2 g/L (triangles), but for 0 and 5 g/L [Bir] the equilibrium seems to last longer. Could the authors find an explanation for this phenomenon?

3) In figure 2 -  the gray lines for hausmannite cannot be discerned (may be the authors should choose another color)

4) Reference no 7 is missing from the reference list (only the number is there)

Reviewer 3 Report

I read this submission several times, in principle, I think it may be recommended for publication after a minor revison.

I suggest these authors to remove some unnecessary contents, which makes this paper so lengthy.

I want to see the authors to explain their work via electronegativity scales, such as ionic electronegativity scale, aqueous electronegativity scale, which make their conclusions more easy to understand. For example, Solution-Phase Electronegativity Scale: Insight into the Chemical Behaviors of Metal Ions in Solution. The Journal of Physical Chemistry A 2012, 116 (16) , 4192-4198. DOI: 10.1021/jp300603f.

For example, the following contents may use the view of ionic electronegativity scales, e.g., Estimation of Electronegativity Values of Elements in Different Valence States. The Journal of Physical Chemistry A , 110(39):11332–11337, oct 2006. "The progressive increase of the cell parameter a proportionally to the initial birnessite amount can be related to a partial substitution of some of the Fe species by Mn species inside the metallic brucite-type sheet of green rust. Indeed, the effective high spin ionic radius of both Mn(II) and Mn(III) (82 and 65 pm) in oxides is higher than the same radius for Fe(II) and Fe(III) species (63 and 49 pm). Such a difference may explain the observed global expansion of the cell. The interlayer distance c depends strongly on the size of the sulfated species that form a double layer [18] and on the electrostatic charge of the layers. However, the c parameter is not very sensitive to the chemical nature of the cations."
